# Activity and Ca²⁺ regulate the mobility of TRPV1 channels in the plasma membrane of sensory neurons

**Eric N Senning, Sharona E Gordon\***

Department of Physiology and Biophysics, University of Washington, Seattle, United States

**Abstract** TRPV1 channels are gated by a variety of thermal, chemical, and mechanical stimuli. We used optical recording of Ca²⁺ influx through TRPV1 to measure activity and mobility of single TRPV1 molecules in isolated dorsal root ganglion neurons and cell lines. The opening of single TRPV1 channels produced sparklets, representing localized regions of elevated Ca²⁺. Unlike sparklets reported for L-type Ca²⁺ channels, TRPV4 channels, and AchR channels, TRPV1 channels diffused laterally in the plasma membrane as they gated. Mobility was highly variable from channel-to-channel and, to a smaller extent, from cell to cell. Most surprisingly, we found that mobility decreased upon channel activation by capsaicin, but only in the presence of extracellular Ca²⁺. We propose that decreased mobility of open TRPV1 could act as a diffusion trap to concentrate channels in cell regions with high activity.

## Introduction

The transient receptor potential vanilloid type 1 (TRPV1) ion channel expressed in peripheral sensory neurons may be one of the most heavily regulated ion channels known. Its gating is multimodal, regulated by extracellular protons, phosphorylation by PKC, phosphorylation by PKA, temperature, fatty acids such as lysophosphatidic acid and anandamide, phosphoinositide signaling lipids, eicosanoids, and capsaicin, the pungent extract of hot chili peppers (*Caterina et al., 1997*; *Zygmunt et al., 1999*; *Premkumar and Ahern, 2000*; *Bhave et al., 2002*; *Hu et al., 2002*; *Pomonis et al., 2003*; *Klein et al., 2008*; *Ufret-Vincenty et al., 2011*; *Nieto-Posadas et al., 2012*). Furthermore, it is also subject to indirect regulation by G-protein coupled receptors and intracellular Ca²⁺ and its trafficking can be regulated by receptor tyrosine kinases (*Chuang et al., 2001*; *Zhang et al., 2005*, *2008*; *Stein et al., 2006*).

TRPV1 channels are highly permeable to Ca²⁺, with $P_{Na}$:$P_{Ca}$ estimated to be at least 1:10 (*Caterina et al., 1997*). Indeed, Ca²⁺ imaging at the whole-cell level was used in the seminal *tour-de-force* that first identified TRPV1 as the capsaicin-activated ion channel (*Caterina et al., 1997*). The high permeability of TRPV1 to Ca²⁺ has also been a useful tool in high-throughput screening of regulatory compounds and led to the identification of a family of toxins, first purified from spiders, that act as potent activators of TRPV1 (and have enhanced the survival of the spiders) (*Caterina et al., 1997*). In addition, Ca²⁺ influx through TRPV1 desensitizes sensory neurons (*Cholewinski et al., 1993*; *Koplas et al., 1997*; *Rosenbaum et al., 2004*). Although multiple pathways are likely involved in neuronal desensitization, depletion of the signaling lipid phosphoinositide 4,5-bisphosphate (PI(4,5)P₂) via Ca²⁺-mediated activation of phospholipase C appears to contribute to desensitization of TRPV1 during periods of high channel activity (*Stein et al., 2006*; *Lukacs et al., 2007*).

Optical recording of localized Ca²⁺ influx through plasma membrane ion channels can be achieved using a combination of Ca²⁺-sensitive fluorescent dyes and non-fluorescent Ca²⁺ chelators loaded into cells via a whole-cell patch pipette. When Ca²⁺-permeable channels open, localized Ca²⁺ influx

**\*For correspondence:** seg@u.
washington.edu

**Competing interests:** The authors declare that no competing interests exist.

**eLife digest** Cells rely on proteins called receptors to keep them informed about what is going on around them. These receptors, which are embedded in the surface of the cell, detect and respond to specific chemical signals. It is known that receptors move around the cell surface as they search for particular chemical signals, but these movements have not been widely studied in experiments.

Senning and Gordon have now investigated the movements of receptors called TRPV1 channels that can detect a chemical called capsaicin. This receptor contains an ion channel that is usually closed. However, when the receptor is activated this channel opens and allows calcium ions to enter the cell.

In the experiments the receptors were tagged with a fluorescent marker, and a fluorescent calcium dye was used to indicate when the channel had been activated by capsaicin. This allowed the function of the receptors to be followed in real time. The experiments were performed on nerve cells taken from mice and in cell culture lines derived from neurons and kidney cells.

Senning and Gordon showed that at first the receptors moved around freely on the surface of the cell, with the degree of mobility varying from cell to cell and also from receptor to receptor. However, when a receptor detected a capsaicin molecule and opened, it tended to move more slowly when calcium ions were present outside the cells.

Further research is needed to explore the mechanism that prevents the receptor from moving. However, since calcium ions are involved in a wide range of processes in the nervous system, it is thought that this mechanism ensures that the receptors concentrate in regions of high neuronal activity.

produces a fluorescent 'sparklet' in the cytosol proximal to the active channel (*Wang et al., 2001*). The presence of the nonfluorescent $Ca^{2+}$ chelator in the cell acts as a sink for the excess $Ca^{2+}$, enhancing the localization of the source of the influx (*Navedo et al., 2005*). Optical approaches have been used to record the activity of L-type $Ca^{2+}$ channels in urinary bladder smooth muscle (*Sidaway and Teramoto, 2014*), arterial smooth muscle (*Navedo et al., 2006*; *Amberg et al., 2007*; *Navedo et al., 2010*; *Tajada et al., 2013*), ventricular myocytes (*Wang et al., 2001*; *Zhou et al., 2009*), and mammalian cell lines (*Gulia et al., 2013*). More recently, sparklets due to TRPV4 channels have been reported in arterial smooth muscle (*Mercado et al., 2014*) and vascular endothelium (*Bagher et al., 2012*; *Sonkusare et al., 2012*).

Two aspects of sparklets reported from L-type $Ca^{2+}$ channels and TRPV4 channels are remarkable. First, multiple channels were typically clustered at the sparklet sites. Second, the sparklets remained stationary throughout the observation period. Thus, some mechanism(s) for clustering channels must be operating in these cells. Whether the clustering mechanism(s) and the mechanism(s) eliminating diffusion of the clusters are related is unknown. Most importantly, whether any $Ca^{2+}$-permeable channels have the capability to gate (open and close) as they diffuse laterally in the plasma membrane of a cell has not previously been addressed. It should be noted that the muscle nicotinic aceytylcholine receptors (AChR) expressed in *Xenopus* oocytes have also been studied by optical recording, and the fluorescence signals emanating from these channels did not indicate channel clustering at the fluorescence sites (*Demuro and Parker, 2005*). Nevertheless, the authors did find that all fluorescence $Ca^{2+}$ signals from AChR maintained a constant position for the duration of the optical recordings.

Regulation of mobility in the plasma membrane has been identified as a key element in signaling for the Orai family of $Ca^{2+}$-release activated channels (CRAC). Orai channels diffuse throughout the plasma membrane in resting cells, but in response to the emptying of $Ca^{2+}$ from the endoplasmic reticulum (ER) they cluster at sites in the surface membrane that juxtapose to the ER (*Lioudyno et al., 2008*; *Penna et al., 2008*). The interaction of Orai channels with the ER-resident protein STIM1 reduces Orai mobility, acting as a sort of diffusion trap to localize Orai channels to these sites as well as directly gating $Ca^{2+}$ influx through the Orai pore (*Yeromin et al., 2006*; *Zhang et al., 2006*; *Wu et al., 2014*). Although the diffusion trap mechanism has not yet been proposed for other types of ion channels, the addition of regulated mobility to a cell's toolkit for controlling its functions represents a powerful means of increasing the spatial and temporal specificity of cell signaling.

In the present study we asked whether the mobility of TRPV1 might be regulated and whether any such regulation might be coupled to channel activity. We took advantage of the high $Ca^{2+}$ permeability of TRPV1 to record $Ca^{2+}$ sparklets that reflected the influx of $Ca^{2+}$ through open TRPV1 channels in response to capsaicin in isolated mouse dorsal root ganglion neurons and in immortalized mammalian cell lines. Whole-cell voltage clamp was used to both minimize the signal due to voltage-gated $Ca^{2+}$ channels and to introduce a combination of fluorescent and nonfluorescent $Ca^{2+}$ chelators. In contrast to L-type $Ca^{2+}$ channels, TRPV4 channels and AChR previously studied, $Ca^{2+}$ sparklets from TRPV1 channels in isolated dorsal root ganglion neurons and cultured cell lines were observed with a wide spectrum of mobilities, from immobile to near the diffusion coefficient of a lipid moving through a bilayer. To our knowledge, the opening and closing of a channel as it diffuses through the plasma membrane has not been previously observed. Using TRPV1 fused to GFP expressed in HEK293T/17 cells, we established that each sparklet represented the activity of one and only one TRPV1 channel. Remarkably, we found that the mobility of TRPV1, whether measured using sparklets or by tracking GFP-fused channels, decreased with activity in an extracellular $Ca^{2+}$-dependent manner. We propose that activity-dependent regulation of TRPV1 mobility could, like for Orai, lead to clustering of channels in regions of the cell where they are most needed.

## Results

The intricate regulation of TRPV1 *activity* prompted us to examine whether TRPV1 *mobility* would also be subject to regulation. This question arose, in part, from the recognition that the many signals converging on TRPV1-expressing sensory neurons would be distributed heterogeneously across the cells' receptive fields. In addition, new data from other channels indicates that mobility can be a powerful means by which cell function can be regulated. To examine this question, however, required implementation of new approaches that would allow activity and mobility to be measured in the same preparation and in primary sensory neurons.

### Capsaicin-activated sparklets in isolated dorsal root ganglion neurons

To determine the activity of individual TRPV1 channels, we used whole-cell patch clamp of isolated mouse dorsal root ganglion neurons. TRPV1-expressing nociceptors were easily distinguished as the small-diameter (~50 μm) cells in a 12–24 hr culture. As previously shown, these cells have large, capsaicin-activated currents when studied with whole-cell patch clamp as well as when studied using inside-out patch clamp (*Stein et al., 2006*). The fluorescent $Ca^{2+}$ indicator Fluo-4 (200 μM) and the nonfluorescent $Ca^{2+}$ chelator EGTA (10 mM) were included in the patch pipette, and cells were held at a potential of −40 mV. This potential was chosen empirically as the potential that minimized $Ca^{2+}$ influx in the absence of capsaicin. In addition to the EGTA included in the intracellular pipette solution, 1 μM thapsigargin was included in the bath solution to deplete the endoplasmic reticulum $Ca^{2+}$ stores (see 'Materials and methods'). These solutions allowed us to attribute the capsaicin-induced rise in intracellular $Ca^{2+}$ to influx through TRPV1 channels in the plasma membrane.

Very little fluorescence was observed upon dialysis of Fluo-4/ EGTA solution into the isolated dorsal root ganglion neurons via the patch pipette. However, subsequent perfusion of 100 nM capsaicin into the bath generated localized bursts of fluorescence, termed sparklets (*Figure 1A* and *Video 1*). Sparklets were observed with physiological concentrations of extracellular $Ca^{2+}$ (2 mM). Out of 12 cells that responded to capsaicin and showed a rise in global fluorescence due to $Ca^{2+}$ influx, nine cells had easily distinguished local micro-domains of $Ca^{2+}$ influx that we classified as sparklets.

### Two-state intensity of sparklets

Fluorescence intensity analysis of TRPV1 sparklets in isolated dorsal root ganglion neurons revealed two states: a dark state presumed to correspond to a closed channel state and a bright state presumed to correspond an open channel state (*Figure 1A* and *Video 1*). All-points amplitude histograms of the fluorescence intensity revealed that sparklet-to-sparklet variability in amplitude of the capsaicin-induced fluorescence was low (*Figure 1—figure supplement 1*). However, this amplitude is likely determined primarily by the channel's relative position along the z-axis within the TIRF evanescent excitation field as well as the concentrations and properties of $Ca^{2+}$ buffers used. Thus, we did not interpret the stereotyped amplitude in terms of $Ca^{2+}$ concentration at the sparklet sites. Interestingly, the two-state intensity traces (*Figure 1A* and *Figure 1—figure supplement 1*) resemble those of single-channel activity measured with electrophysiology (e.g., *Liu et al., 2004*; *Rosenbaum et al., 2004*),

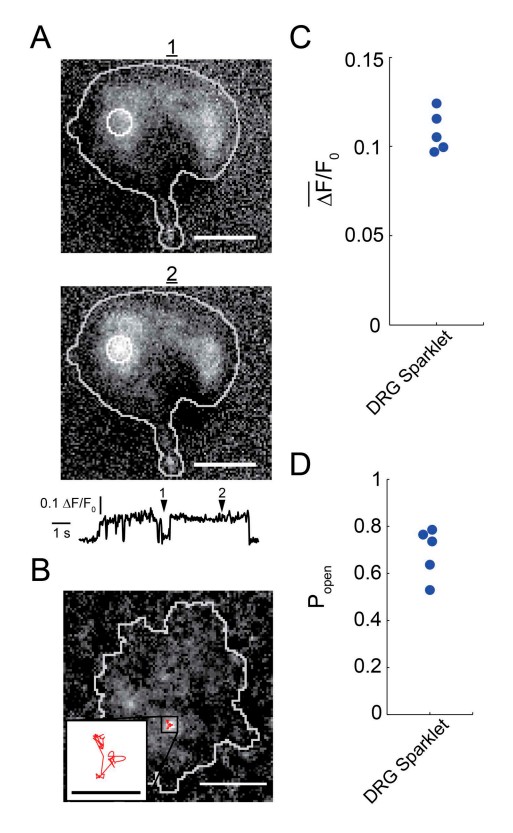

**Figure 1**. Capsaicin induced sparklets in whole-cell voltage clamped dorsal root ganglion cells loaded with Fluo-4. (**A**) Images of immobile sparklet (white circle) in dorsal root ganglion cell at time points when inactive (1) and active (2) with corresponding intensity trace below (see **Video 1**). (**B**) Moving sparklet recorded for 5.6 s in dorsal root ganglion cell (see **Video 2**). Inset is of sparklet trace magnified fivefold. Inset scale bar is 1 μm. All other scale bars are 5 μm. (**C**) Mean fluorescence intensity ($\overline{\Delta F / F_0}$) and (**D**) open probability ($P_{open}$) of dorsal root ganglion sparklets (N = 5; 3 cells) in *Figure 1—figure supplement 1 A–E*. Cut-off for 'open' is when intensity ($\Delta F/F_0$) exceeds 0.05.

The following figure supplements are available for figure 1:

**Figure supplement 1**. All points histogram of immobile sparklets in dorsal root ganglion cells (N = 5).

**Figure supplement 2**. Dose-response relation for activation TRPV1 channels in isolated dorsal root ganglion neurons by capsaicin.

**Figure supplement 3**. Increase in fluorescence intensity due to $Ca^{2+}$ influx into F11 cell transfected with TRPV1-tagRFP, with Fluo-4 introduced via the whole-cell patch pipette.

albeit with lower temporal resolution. These data are consistent with each sparklet representing a single gating TRPV1 channel or a cluster of TRPV1 channels gating cooperatively. We distinguish between these possibilities below.

The all-points histograms of sparklet amplitude shown in *Figure 1—figure supplement 1* could be used to calculate the open probability for each sparklet, shown in *Figure 1D*. We compared these open probability values to those measured from the whole population of channels in these isolated dorsal root ganglion cells using whole-cell patch clamp. Examining the dose–response relation for activation by capsaicin (*Figure 1—figure supplement 2*) shows that the current recorded in the presence of 100 nM capsaicin normalized to the current at a saturating capsaicin concentration was 0.84 ± 0.05 (n = 4). This fractional activation value is slightly greater than the open probability values plotted in *Figure 1D*, for which the mean was 0.7 ± 0.1 (N = 5). This small discrepancy is expected, because the open probability of TRPV1 in the presence of a saturating capsaicin concentration is somewhat less than 1.0 (*Liu et al., 2004*). These data indicate that the sensitivity of single TRPV1 channels as calculated from optical recordings of sparklets is similar to that of the cell population measured using whole-cell patch clamp.

## Sparklets are dynamic, moving laterally in the membrane

The fluorescence intensity of the sparklet shown in *Figure 1A* was straight forward to quantify because it was immobile, that is it did not move. However, many of the capsaicin-activated sparklets we observed were dynamic, that is they appeared to move laterally in the plane of the membrane (*Figure 1B*). In order to follow the high mobility of these sparklets, we recorded their movement at a camera frame-rate of 33 fps. The inset of *Figure 1B* was made by tracking the indicated sparklet with the Spot Tracker 2D plugin for ImageJ (see 'Materials and methods'), and it illustrates the significant mobility of the sparklet site over the course of 5.6 s (see also *Video 2*; further examples of mobile sparklets in *Video 3*). To our knowledge, simultaneous mobility in the plasma membrane and gating has not been previously reported for any ion channel, and are explored further below.

## Neuronal processes running underneath the cell body obscure sparklets

The heterogeneity of TRPV1 mobilities, with some sparklets immobile and others dynamic, indicates

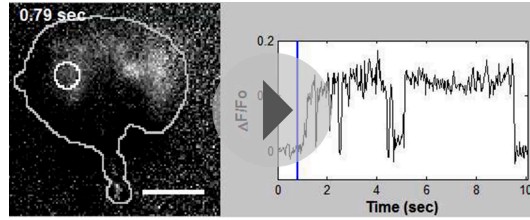

**Video 1**. Immobile sparklet (white circle) in dorsal root ganglion cell with fluorescence recording on the right elicited by capsaicin (100 nM). Scale bar is 5 μm.

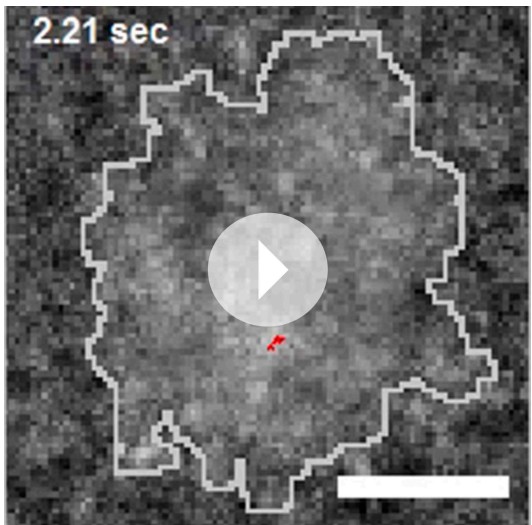

**Video 2**. Moving sparklet observed in whole cell patch clamped dorsal root ganglion cell after addition of capsaicin (100 nM). Fluo-4 is loaded into cell as an indicator for $Ca^{2+}$ influx. Track of sparklet is superimposed in red. Scale bar is 5 μm.

that the population of TRPV1 channels was heterogeneous in some aspect of channel structure or function. To determine the mechanisms involved in controlling TRPV1 mobility we measured the sparklet trajectories, as shown in *Figure 1B*, inset. Two factors, however, limited our ability to fully characterize these trajectories. The first factor, which is a general limitation of our approach in any cell type, was that our sparklet recording time was limited by the properties of the $Ca^{2+}$-sensitive dye used. We required a dye sensitive enough to show the $Ca^{2+}$ flux through just a single channel. The appearance of individual sparklets when capsaicin was added became difficult to discern over time as the fluorescence across the cell footprint increased. The increase in fluorescence 'background' was due to the rapid, global increase in intracellular $Ca^{2+}$ that occurs over the course of our experiments. Although the $Ca^{2+}$ chelator EGTA (10 mM) in our intracellular solution extended the useful recording time by acting as a $Ca^{2+}$ sink, a rise in global $Ca^{2+}$ due to continued influx of $Ca^{2+}$ through TRPV1 ultimately obscured the sparklet signals once intracellular $Ca^{2+}$ levels exceeded the upper limit of the $Ca^{2+}$ sensitive dye's optimal range (Fluo-4: 1 μM $Ca^{2+}$; Life Technologies, Carlsbad, CA).

The second factor limiting our interpretation of sparklet activity in dorsal root ganglion neurons are the tube-like structures prevalent within the footprint of isolated neurons that brightened markedly when capsaicin was added. These tube-like structures were observed even when the SERCA inhibitor thapsigargin was present (*Video 3*), indicating that they are likely neurite-like processes underneath the cell footprint, rather than endoplasmic reticulum within the cell body. The processes hampered further characterization of capsaicin-activated sparklet activity in isolated dorsal root ganglion cells because in the confined space of the processes, global $Ca^{2+}$ levels dominate sparklet signals very rapidly (*Video 3*). We attempted to circumvent this issue by conducting experiments very soon after cell isolation, before processes could form. However, cells studied soon after isolation, before significant processes had formed, showed footprints that were irregularly-adhered to the coverglass, and thus were not evenly within the evanescent TIRF field. Transient transfection of TRPV1 into the F11 cell line, a hybridoma formed between dorsal root ganglion cells and neuroblastoma cells (*Stein et al., 2006*), were examined as an alternative to primary cell culture. In experiments using the F11 line with the channel expressed as a fluorescent fusion protein, TRPV1-tagRFP, we did observe moving sparklets (*Video 4*). However, processes running under the footprint of F11 cells (*Figure 1—figure supplement 3* and *Video 5*) interfered with sparklet characterization as they did in isolated dorsal root ganglion neurons. To study sparklet mobility we therefore turned to expressing tagRFP- and GFP-tagged TRPV1 (TRPV1-tagRFP and TRPV1-GFP) in transiently transfected HEK293T/17 cells, in which the neurite-like processes were not observed.

## TRPV1-tagRFP channels in transiently transfected HEK293T cells show capsaicin-activated $Ca^{2+}$ sparklets

We examined whether HEK293T/17 cells would prove suitable for quantifying TRPV1 mobility and activity. TRPV1 channels expressed robustly in these cells, which proved a problem as single

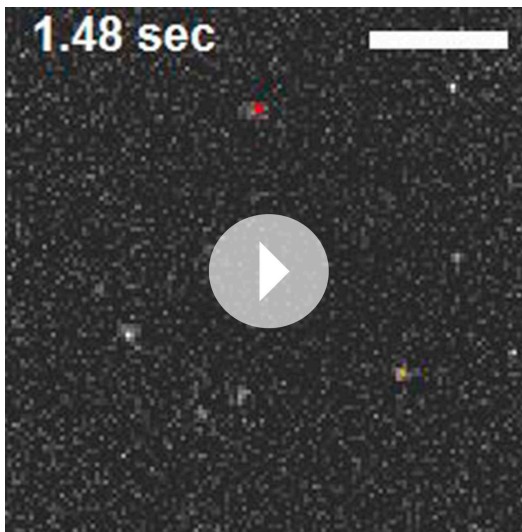

**Video 3**. Numerous mobile sparklets in whole cell patch clamped dorsal root ganglion cell after addition of capsaicin (100 nM). Scale bar is 5 μm.

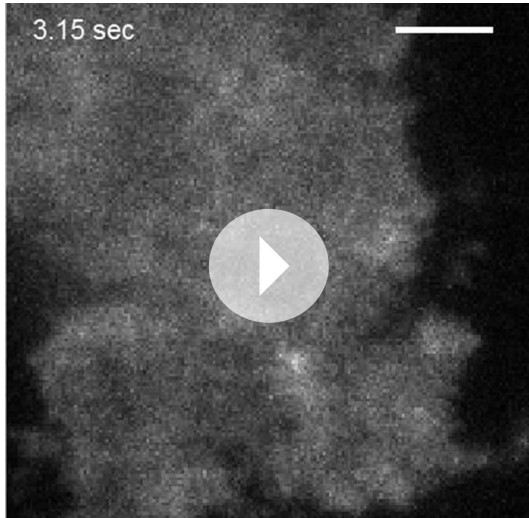

**Video 4**. Immobile and mobile sparklets in TRPV1-tagRFP transfected F11 cell after addition of capsaicin (100 nM). Arrow indicates the site where a moving sparklet appears in a process at 7.75 s. This video is an exception in that the frame rate is 20 Hz. Scale bar is 5 μm.

tagRFP-tagged channels could not easily be distinguished and therefore nor would individual sparklets. However, we found that performing our experiments within 12–18 hr of transfection yielded the sparse density of channels required.

We found that, like isolated dorsal root ganglion neurons, TRPV1-tagRFP and TRPV1-GFP in HEK293T/17 cells showed mobile and immobile capsaicin-activated sparklets (*Video 6*). Sparklets were not observed in the absence of capsaicin nor in the absence of extracellular $Ca^{2+}$ (*Figure 2A*). As shown in *Figure 2B,C* (see also *Video 7*), fluorescence from capsaicin-activated sparklets in HEK293T/17 cells had the signature two-amplitude intensities also observed in isolated dorsal root ganglion neurons (*Figure 1—figure supplement 1*). For these experiments we used Fluo-5F (200 μM) together with 10 mM EGTA in the pipette solution and 20 mM $Ca^{2+}$ in the extracellular buffer to closely mimic the experiments of *Navedo et al. (2005)*, who report on immobile sparklets from L-type $Ca^{2+}$ channels (see 'Materials and methods'). Interestingly, the amplitude of fluorescence intensity in the sparklets was comparable in isolated dorsal root ganglion neurons and HEK293T/17 cells (compare *Figure 2D* to *Figure 1C*). However, differences in the dye (Fluo-5F vs Fluo-4) and extracellular $Ca^{2+}$ concentrations (20 mM vs 2 mM) make it difficult to draw any conclusions from this similarity. The open probability for sparklets in HEK293T/17 cells was significantly less than for sparklets in isolated dorsal root ganglion neurons (*Figure 2E*). This difference is consistent with the higher $K_{1/2}$ for activation of TRPV1 expressed in HEK293T/17 cells (*Collins and Gordon, 2013*). Interestingly, capsaicin-independent increases in local $Ca^{2+}$ were observed in untransfected as well as TRPV1-transfected HEK293T/17 cells. These could be easily distinguished from sparklets due to TRPV1 as their open times were very brief and their amplitude was spikey (*Figure 2—figure supplement 1*), rather than showing two states like TRPV1 (*Figure 2B,C*). These TRPV1-independent $Ca^{2+}$ events were excluded from the analysis reported here.

## Each fluorescent spot represents one and only one TRPV1 channel

The two-state fluorescence intensity we observed with TIRF imaging of capsaicin-induced $Ca^{2+}$ sparklets in dorsal root ganglion neurons and HEK293T/17 cells suggested that each sparklet was due to a single TRPV1 channel. To test this hypothesis, we used photobleaching to count the number of subunits per feature, using GFP instead of tagRFP due to its superior spectroscopic properties and because we did not need to avoid spectral overlap with the $Ca^{2+}$-sensitive dye in these experiments. TRPV1 is a homomeric tetramer structurally similar to the voltage-gated superfamily of ion channels that includes voltage-gated $K^+$ channels (*Liao et al., 2013*). Each TRPV1-GFP channel should thus have

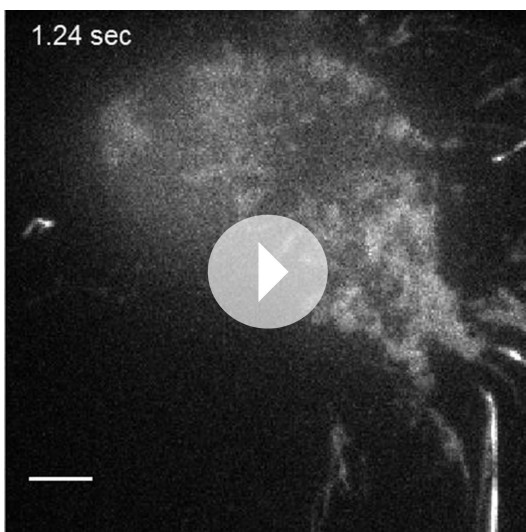

**Video 5**. Sparklets occurring in TRPV1-tagRFP transfected F11 cell after addition of capsaicin (100 nM). Pinpoint accuracy of sparklets is complicated by the uneven cell footprint within the TIRF evanescent field. Scale bar is 5 µm. See **Figure 1—figure supplement 2**.

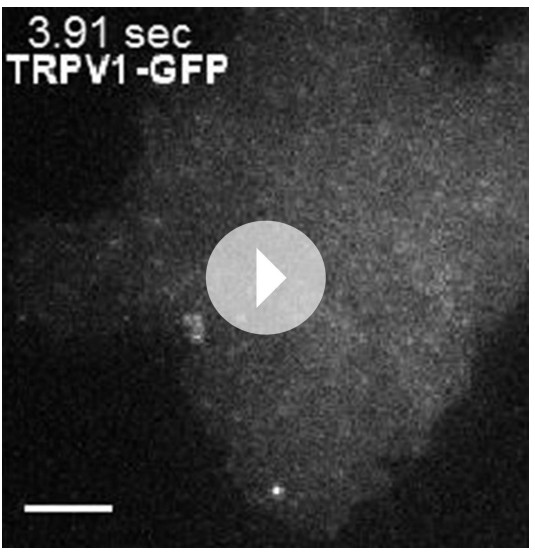

**Video 6**. Immobile and mobile sparklets in TRPV1-GFP transfected HEK293T/17 cell after addition of capsaicin (100 nM). TRPV1-GFP fluorescence is observed in the first part of the video, and fluorescent sparklets become the dominant signal in second half of video with both mobile (bottom middle) and immobile sparklets present. Scale bar is 5 µm.

four copies of GFP per functional channel. If each fluorescent spot represents one TRPV1-GFP channel, four photobleaching steps should be observed. However, if each spot represents several TRPV1-GFP channels, >4 photobleaching steps should be observed. Because measuring the GFP fluorescence intensity of mobile TRPV1 channels proved difficult, paraformaldehyde-fixed cells were used in these photobleaching experiments.

As shown in **Figure 3A**, using a high-intensity laser to image the fluorescence from individual TRPV1-GFP features produced sudden decreases in fluorescence intensity, each of which is presumed to represent photobleaching of a single GFP (**Ulbrich and Isacoff, 2007**). We observed from one to four photobleaching steps across the population of TRPV1-GFP and plotted a histogram of the steps (total number of sites with bleaching steps = 384), shown in **Figure 3B**. We then fitted the data with a binomial model given a fixed number of subunits, $n$, and the probability that each GFP monomer was able to fluoresce at the time the sample was first illuminated, $p$, as a free parameter (see 'Material and methods'). The best fit was obtained using $n = 4$ and $p = 0.45$ (**Figure 3B**), and fits with $n = 5$, as would be the case if TRPV1 were pentameric, or $n = 8$, if TRPV1 tetrameric channels formed pairs, were both rejected by $\chi^2$ goodness-of-fit tests (**Figure 3C** and legend). Although $p = 0.45$ produced by the fit with $n = 4$ is low with respect to this value reported by others (**Ulbrich and Isacoff, 2007**; **Ji et al., 2008**; **Zhang et al., 2009**), our experiments were performed in fixed cells, a preparation known to partially disrupt fluorescent protein fluorescence (**Chalfie and Kain, 1998**). Together with the two-state fluorescence intensity of capsaicin-induced sparklets, these data indicate that each TRPV1-GFP feature represented one channel, with four subunits, rather than two or more channels, with eight or more subunits.

## Mobility of TRPV1-GFP expressed in HEK293T/17 cells

Unlike sparklet experiments, in which the duration of recording was limited by the properties of the Ca²⁺ chelators used, TRPV1-GFP experiments could proceed considerably longer. To further understand mechanisms underlying the spectrum of sparklet mobilities we observed, we performed a large survey of TRPV1-GFP mobilities by tracking diffusion of the channels in the plasma membrane. We acquired 15 s videos of TRPV1-GFP channels moving laterally in the plasma membrane of living cells and used the particle tracking algorithm u-track, which simplifies tracking of features in a large sample size and minimizes user bias in track identification (**Jaqaman et al., 2008**), to follow channel movements. Channel trajectories for the

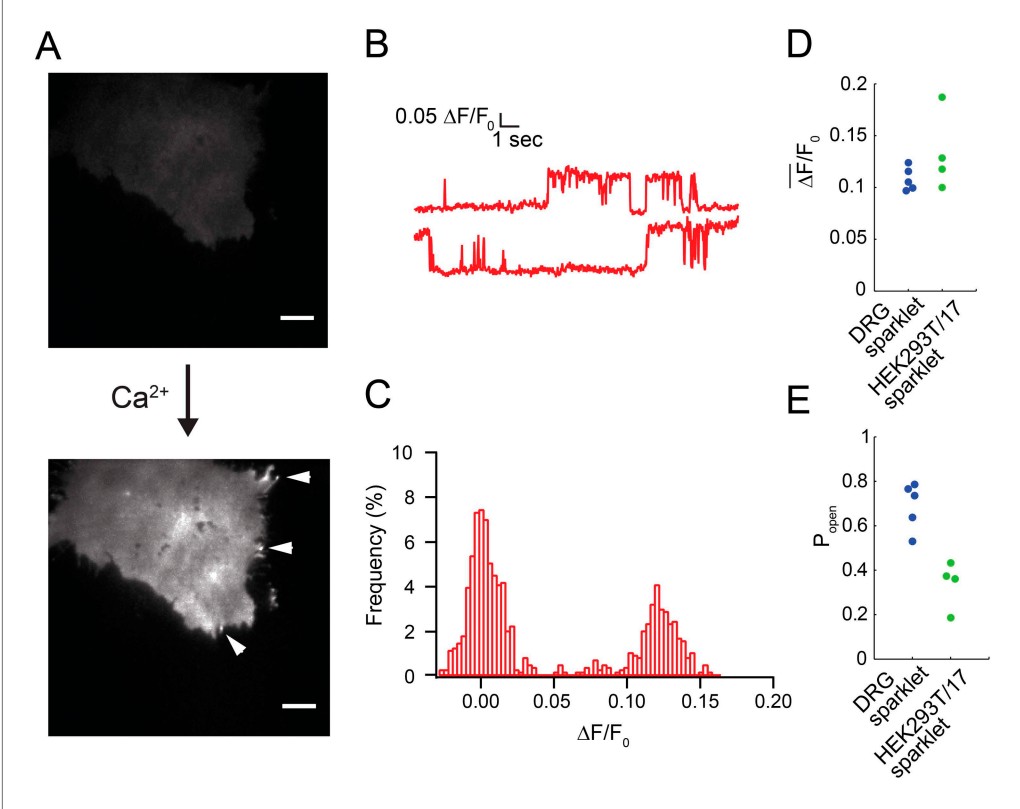

**Figure 2**. Fluorescence intensity traces of immobile sparklet sites in cells loaded with Fluo-5F and extracellular buffer containing 20 mM $Ca^{2+}$ and voltage-clamped as described in 'Materials and methods'. (**A**) Without $Ca^{2+}$ in the extracellular buffer very little green fluorescence was observed in the capsaicin-treated (100 nM) HEK293T/17 cell that was transiently transfected with TRPV1-tagRFP. The image in the top panel was acquired after a 30 s exposure to capsaicin. Addition of 20 mM $Ca^{2+}$ to the extracellular buffer increased the fluorescence of the cell footprint dramatically and individual sparklet sites could be observed (indicated with white arrowheads in bottom panel). TIRF images have matching lookup table (LUT) values and have not been otherwise altered. Scale bar is 5 µm. (**B**) Representative trace of sparklet from TRPV1-tagRFP expressing HEK293T/17 cell. (**C**) All points histogram of trace in (**B**). (**D**) Mean fluorescence intensity ($\overline{\Delta F/F_0}$) and (**E**) open probability ($P_{open}$) of dorsal root ganglion sparklets (Blue; same data as in *Figure 1C,D*) and TRPV1-tagRFP sparklets in HEK293T/17 cells (green; N = 4). Cut-off for 'open' is when intensity ($\Delta F/F_0$) exceeded 0.05. See *Video 7*.

The following figure supplement is available for figure 2:

**Figure supplement 1**. Fluorescence intensity traces of TRPV1-independent $Ca^{2+}$ events in HEK293T/17 cells loaded with Fluo-5F, with extracellular buffer containing 20 mM $Ca^{2+}$, and studied with whole-cell voltage clamp as described in 'Materials and methods'.

first 6 s of a video are shown projected onto the last fluorescent image of the cell in *Figure 4A* (see *Video 8*). The mean squared displacements (MSD) of selected tracks labeled as 1, 2, and 3 are presented in *Figure 4B*. Each of the three tracks exhibited a different MSD vs time slope, which was characterized by an effective diffusion coefficient, $D_{eff}$, as a measure of mobility. These data were then used to generate a histogram of effective diffusion coefficients for all the TRPV1-GFP channel tracks observed in the full 15 s video (*Figure 4C*). The large variability in channel mobilities, with a 20-fold range of $D_{eff}$ (0.01–0.5 µm²/s), is consistent with a heterogeneous population within any given cell. Interestingly, the distribution of mobilities between cells also varied, but only by about a factor of two (mean $D_{eff}$ range: 0.029 to 0.056 µm²/s) (*Figure 4D* and *Figure 4—figure supplement 1*).

## Mobile sparklets mediated by TRPV1-GFP

To establish a one-to-one correspondence between capsaicin-induced sparklets and mobile TRPV1-GFP, we implemented an optical method to coincidentally detect sparklets while simultaneously

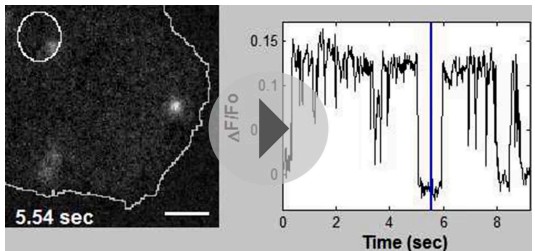

**Video 7.** Immobile sparklet used to analyze TRPV1-tagRFP sparklet intensity levels (See **Figure 2B**). Scale bar is 5 μm.

tracking the channel movement with a GFP label. **Figure 5A** (red trace) shows the trajectory of a typical TRPV1-GFP channel that emitted sparklets along its track. In **Figure 5B** we plot the fluorescence intensity of the moving TRPV1-GFP channel as it increased intermittently to higher levels (see also **Videos 9 and 10**).

To compare the confidence with which the sparklets and TRPV1-GFP fluorescence could be localized, we measured the uncertainty of the precision as an average of the standard deviations of a 2D bivariate Gaussian fitted to the fluorescence feature in each step of the trajectory shown in **Figure 5A**. We used circles representing the size of the uncertainty (1/10th scale) to indicate both which points were observed as sparklets (**Figure 5C**, cyan circles) and which as GFP (**Figure 5C**, red circles). The uncertainty for all the points (GFP and sparklet) ranged from 0.13 μm to 0.33 μm, allowing us to determine the positions of the fluorescent feature along the full track length. Two interpretations of these data stand out. First, the co-localization between TRPV1-GFP (**Figure 5C**, red circles) and the $Ca^{2+}$ sparklets (**Figure 5C**, cyan circles) confirmed that mobile sparklets co-diffuse with mobile TRPV1-GFP channels. Second, the higher fluorescence intensity of sparklets compared to TRPV1-GFP did not impede our ability to localize sparklets as compared to TRPV1-GFP. The simplest interpretation of our data is that sparklets represent $Ca^{2+}$ flux through the labeled TRPV1-GFP as it gates. The mobility of actively gating TRPV1 is somewhat surprising, as $Ca^{2+}$ sparklets from TRPV4 were previously reported to be immobile (**Mercado et al., 2014**) as have sparklets attributed to L-type $Ca^{2+}$ channels (**Navedo et al., 2005**; **Navedo et al., 2006**) and AchR channels (**Demuro and Parker, 2005**). In addition, in a search of the literature we did not find any reports of simultaneous observations of ion channel activity and mobility in living cells. However, the mobility of capsaicin-activated TRPV1 sparklets in three different cell types (isolated dorsal root ganglion neurons, F11 cells, and HEK293T/17 cells) raises the question of whether the mobility of other types of active channels should be reexamined.

## Interplay between TRPV1 mobility and function

Does the heterogeneity of effective diffusion coefficients measured for the population of TRPV1-GFP reflect a functional heterogeneity of the channels? Upon examining a large number of mobile TRPV1-GFP channels under various conditions, we noticed that capsaicin appeared to decrease their mobility.

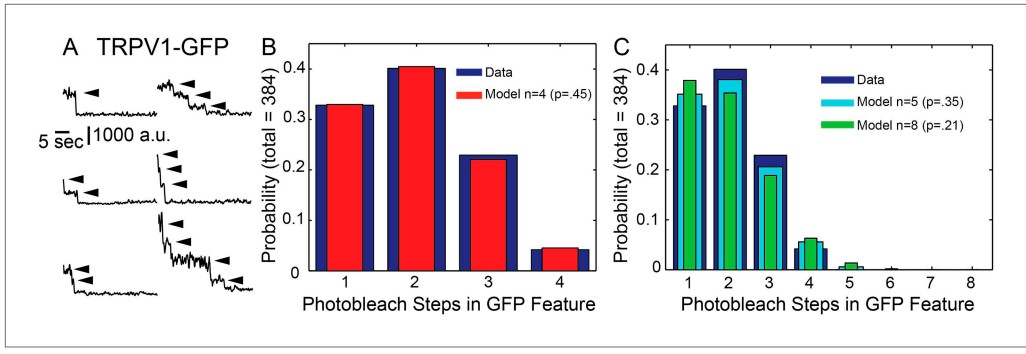

**Figure 3.** Photobleachstep analysis of TRPV1-GFP in fixed HEK293T/17 cells. (**A**) Fluorescence time traces of six representative TRPV1-GFP features. Arrowheads indicate individual photobleach steps (**B**) Histogram of TRPV1-GFP bleach steps in fixed HEK293T/17 cells. TRPV1-GFP fluorescence is characterized as having 1–4 bleaching steps (blue). A zero-truncated binomial distribution fit by maximum likelihood estimate with n = 4 to give a probability of 0.45 (red) ($\chi^2$ goodness-of-fit is 0.1712 and the model is acceptable). (**C**) Fits to zero-truncated binomial distributions with n = 5 (cyan) or n = 8 (green) by maximum likelihood estimate gave probabilities of 0.35 and 0.21, respectively. Chi-squared goodness-of-fit tests were done with n = 5 and n = 8 models, yielding $\chi^2$ values of 4.0717 and 16.386, respectively.

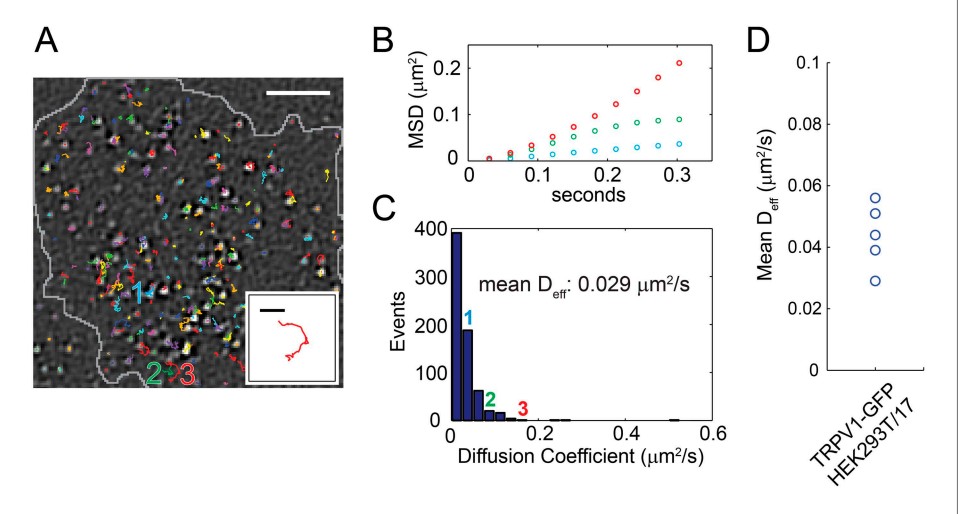

**Figure 4**. Distribution of TRPV1-GFP mobility in HEK293T/17 cell (not voltage-clamped; see 'Materials and methods'). (**A**) TRPV1-GFP channel tracks in a 200 frame video (33 frames per second). Tracks are identified with the u-track algorithm (see 'Materials and methods') and are overlaid on the final image of the series (See *Video 8*). Extent of cell body is shown in grey. The inset is an isolated track from the bottom center of the larger panel labeled '3'. (**B**) Mean square displacements (MSD) at different time lags of three tracks identified as 1 (blue), 2 (green), and 3 (red) in (**A**). (**C**) A Histogram of effective diffusion coefficients derived from tracks in the full 15 s video for the cell shown in (**A**). Tracks identified as 1, 2, and 3 in (**A**) are indicated over their assigned bins. (**D**) Mean effective diffusion coefficients calculated from TRPV1-GFP tracks in HEK293T/17 cells (N = 5). Scale bar of image in (**A**) is 5 µm and inset scale bar is 1 µm.

The following figure supplement is available for figure 4:

**Figure supplement 1**. Distribution of effective diffusion coefficient ($D_{eff}$) in different cells (**A–D**).

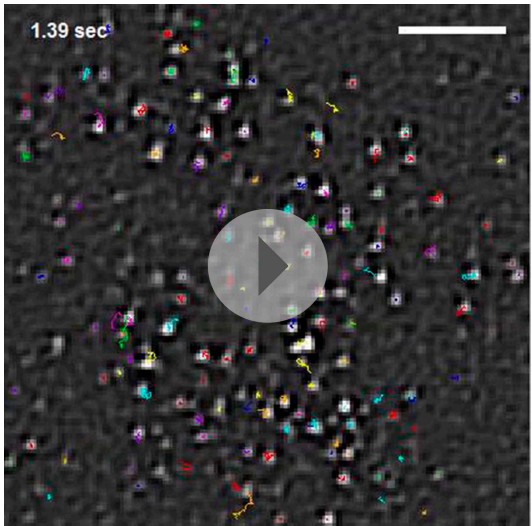

**Video 8**. Video of TRPV1-GFP tracks in transfected HEK293T/17 cell (See *Figure 4*). Scale bar is 5 µm.

We therefore hypothesized that channel mobility might be regulated as a function of channel activity. Because our experiments produced direct measurements of activity and mobility of single TRPV1 molecules, we could determine whether mobility changed upon channel activation.

To test the hypothesis that TRPV1 activity may regulate its mobility, we examined whether the addition of capsaicin to HEK293T/17 cells transiently transfected with TRPV1-GFP produced a change in channel mobility. In control experiments with nominally $Ca^{2+}$-free extracellular buffer and in which the cells were not subject to voltage clamp, we observed a trend of increasing MSD (*Figure 6A*). Although we did not study this effect further, we speculate it was due to photodamage by the excitation light. We did, however, design further experiments to take the observation-induced increase in apparent mobility into consideration. By expressing the data as a ratio of the change in MSD ($R_{\Delta MSD}$, see 'Materials and methods' and *Figure 6—figure supplement 1*) before addition of capsaicin to that after the addition of capsaicin for TRPV1-GFP trajectories within the same cell, we were able to correct for the observation-induced effect on TRPV1-GFP mobility.

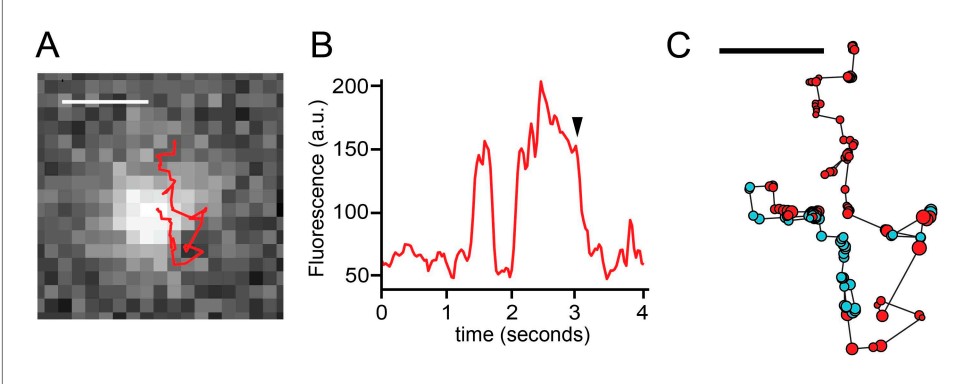

**Figure 5**. TRPV1-GFP and sparklet observed in a moving channel in a voltage-clamped HEK293T/17 cell. (**A**) Track of TRPV1-GFP with increased sparklet intensity. Scale bar is 1 µm. (**B**) Fluorescence intensity trace of mobile TRPV1-GFP channel in (**A**) with sustained higher levels of fluorescence due to sparklet activity. The background image in (**A**) corresponds to the time annotated with an arrowhead in panel (**B**). (**C**) TRPV1-GFP and sparklet positions are shown as filled circles. Red circles are positions assigned to TRPV1-GFP and cyan circles are sparklet positions. Each circle's radius is 1/10 of the standard deviation attributed to the 2-D bivariate Gaussian fitted to the fluorescence feature at that position. Scale bar is 0.5 µm. See *Videos 9 and 10*.

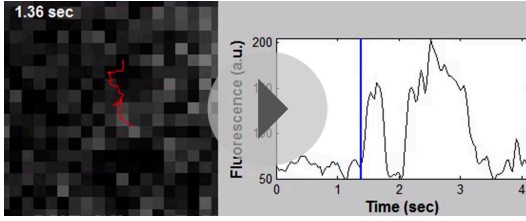

**Video 9**. The TRPV1-GFP channel in the left video can be followed as a GFP feature as of 0.0 s with intermittent sparklet activity and at 6.33 s is a sparklet feature for the majority of the time. Scale is 6.25 pixels per 1 µm.

In cells treated with 1 µM capsaicin and remaining in a $Ca^{2+}$-containing solution (but, as in control experiments, not voltage clamped), the $R_{\Delta MSD}$ of TRPV1-GFP in a given cell decreased upon the addition of capsaicin, indicating that the TRPV1 channels exhibited a decrease in mobility as they opened in the presence of $Ca^{2+}$ (*Figure 6B*, blue). In control experiments that switched from a $Ca^{2+}$-containing solution to a nominally $Ca^{2+}$-free solution without capsaicin (*Figure 6B*, green) or to a nominally $Ca^{2+}$-free solution with capsaicin (*Figure 6B*, red) no significant change between $R_{\Delta MSD}$ in our before and after videos was observed. This difference in $R_{\Delta MSD}$ between before and after capsaicin treatment while in the presence of $Ca^{2+}$ persisted out to 120 msec (*Figure 6—figure supplement 2*). Thus, the decrease in $R_{\Delta MSD}$ required both channel activation (i.e., capsaicin) and $Ca^{2+}$ influx.

Our data indicate that the mobility of TRPV1-GFP decreased in a capsaicin- and $Ca^{2+}$-dependent manner. Regulation of channel mobility by its activity was accessible because: 1) experiments were not truncated by saturation of a $Ca^{2+}$-sensitive dye; 2) the recording of hundreds of channels per cell was possible when tracking mobility with GFP, whereas a much lower channel density was necessary for studying sparklets, 3) the HEK293T/17 cells were not voltage-clamped, allowing higher throughput imaging; and, most importantly, 4) unlike the $Ca^{2+}$ dyes used to image sparklets, the fluorophore used for tracking the channels, GFP, could be imaged even when the channels were closed and even in the absence of extracellular $Ca^{2+}$. The differences in experimental conditions between these experiments and those in which we measured the mobility of sparklets raised the question of whether an activity-dependent decrease in mobility would be observed for capsaicin-activated sparklets.

## Sparklet mobility decreases over time

Sparklets could be imaged only when channels were open and conditions were permissive of $Ca^{2+}$ influx. Thus, the mobility of sparklets due to TRPV1 could not be measured before the addition of capsaicin to the bath. We therefore designed experiments to look at the trend in mobility for individual sparklets that took advantage of the relatively slow perfusion time of our chamber (estimated flow of five-times chamber volume in 30 s) using voltage-clamped HEK293T/17 cells as described in 'Materials and methods'. The open probability of the channels would be expected to increase from near zero in

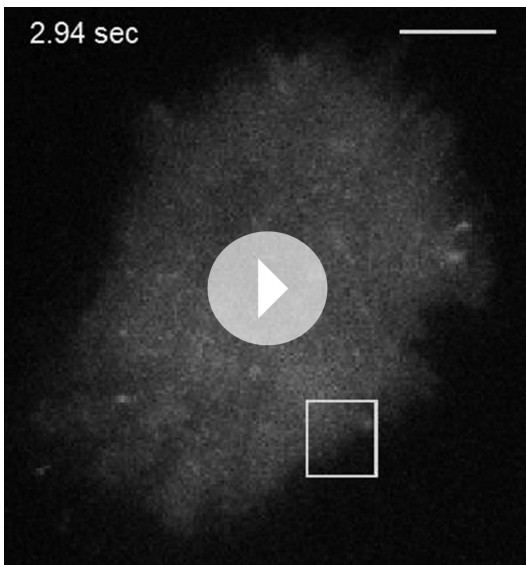

**Video 10**. Video with expanded view of *Figure 5A* and *Video 9*. This video shows the highly dynamic pool of TRPV1-GFP features before capsaicin (100 nM) opens the channels and the GFP signal is overwhelmed by sparklet activity and background Ca²⁺ levels. Box indicates the approximate location of the ROI used for *Figure 5* and *Video 9*. Scale bar is 5 µm.

the absence of capsaicin to ~0.3–0.4 when the bath solution had equilibrated. By measuring the time dependence of mobility under these conditions, we accumulated a set of trajectories on mobile and active TRPV1-GFP or TRPV1-tagRFP channels in which the concentration of capsaicin, and thus their open probability, increased over time. Our hypothesis predicts that the mobility of these sparklets will decrease over the course of the experiment.

We found that 13 out of 16 sparklets analyzed during a period when they remained open and observable showed a capsaicin-dependent decrease in mobility (*Figure 7*). The decreased mobility of these sparklets is illustrated by the trajectory shown in *Figure 7A* (top). The red, blue, and green segments of the track represent the location of the sparklet during three consecutive epochs of equal duration. The first, red epoch was one of high mobility. The second and third epochs, represented in blue and green, respectively, showed markedly reduced mobility. Plotting the displacements as a function of time for this track (*Figure 7A* bottom), fitting it with a line, and calculating its slope was used to measure the trend in this track toward reduced mobility. One of the three sparklets that showed increased mobility upon activation is shown in *Figure 7—*

figure supplement 1. The slopes of the lines from the 16 sparklets are shown in *Figure 7B*, revealing a trend towards decreasing mobility (negative slope; *Figure 7B*). In conclusion, activation of TRPV1 yielded a decrease in channel mobility, whether observed as the GFP-fusion protein or as sparklets.

## Discussion

We recorded the activity of individual TRPV1 channels in live cells by imaging capsaicin-activated optical sparklets with TIRF microscopy. A fluorescent Ca²⁺ indicator (Fluo-4 or Fluo-5F) was introduced, along with the nonfluorescent Ca²⁺ chelater EGTA, into cells via a patch pipette, which also served to clamp the voltage at a potential empirically determined to minimize Ca²⁺ influx in the absence of capsaicin. Capsaicin and extracellular Ca²⁺ were required for sparklet formation in isolated dorsal root ganglion neurons and HEK293T/17 cells transiently transfected with TRPV1-tagRFP. TRPV1 sparklets are a new tool to address critical gaps in our knowledge concerning the function of TRPV1 in its native plasma membrane environment and opens the door to further exploration into how plasma membrane dynamics and cell signaling regulate TRPV1 function.

Synchronous recording of localization and activity of native TRPV1 channels in isolated dorsal root ganglion neurons and of TRPV1-GFP or TRPV1-tagRFP channels in transiently-transfected HEK293T/17 cells revealed a surprising range of mobilities in the plasma membrane. Like the L-type Ca²⁺ channels (*Navedo et al., 2005*; *Navedo et al., 2006*), AchR channels (*Demuro and Parker, 2005*), and TRPV4 channels (*Mercado et al., 2014*) discussed above, some sparklets were immobile, whereas others diffused at a rate comparable to that expected for an individual lipid (~1 µm²/s) (*Golebiewska et al., 2008*) (*Figures 1, 2, 5 and 7*). To our knowledge, gating of channels as they diffuse in the plasma membrane has not been previously reported, although synchronous gating and diffusion may well represent the 'natural' behavior of many channels and transporters.

We combined our studies of capsaicin-induced sparklets in isolated dorsal root ganglion neurons and transiently-transfected HEK293T/17 cells with imaging studies of TRPV1-GFP in HEK293T/17 cells to determine the number of TRPV1 channels underlying the sparklets. Sparklets have been used previously by Navedo et al. to report on the gating of L-type Ca²⁺ channels as single channels or in clusters of up to seven channels (*Navedo et al., 2005*), or assemblies of TRPV4 channels in arterial myocytes

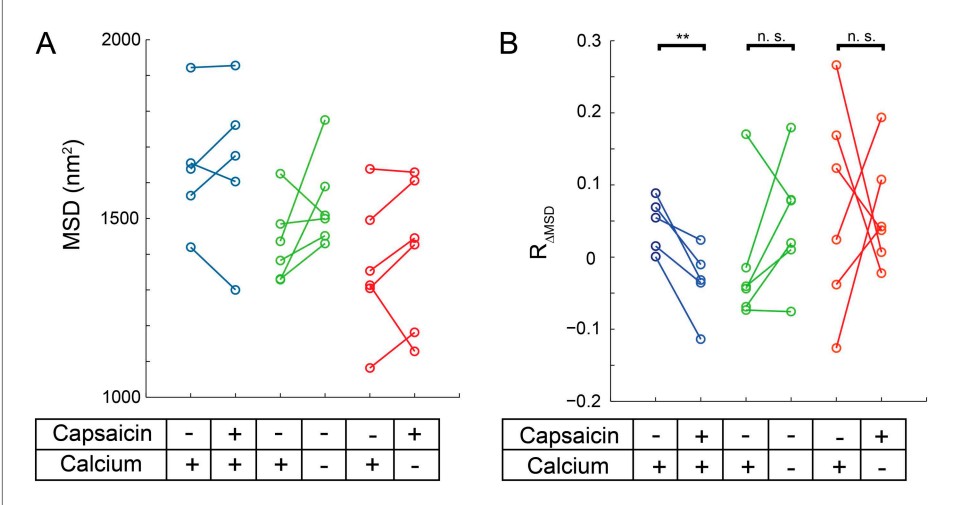

**Figure 6**. Mobility change in TRPV1-GFP channels after capsaicin treatment in cells not subject to voltage clamp. (**A**) HEK293T/17 cells expressing low numbers of TRPV1-GFP are initially imaged for 15 s and then a second 15 s video follows perfusion with a 1 μM capsaicin solution (described in 'Materials and methods'). All three treatment conditions begin in HBSS (containing $Ca^{2+}$) with no capsaicin. Average MSD (time lag of 90 msec) for all tracks shows no significant change in mobility from before to after capsaicin. (**B**) Using the same source data, we characterize mobility with the MSD difference ratio ($R_{\Delta MSD}$, described in 'Materials and methods' and ***Figure 6—figure supplement 1***). With the switch from no capsaicin to 1 μM capsaicin but maintaining the $Ca^{2+}$ concentration at 1.8 mM, the $R_{\Delta MSD}$ drops a significant amount from the initial condition (N = 5, blue data points) as determined by a one-tailed paired t-test (p < 0.01). This significant drop in $R_{\Delta MSD}$ from the initial video to the second video is not observed under conditions where the second solution contains no capsaicin and no added $Ca^{2+}$ (N = 6 cells, green data points; n.s.: not significant) nor is it seen when the second solution contains capsaicin (1 μM) but no added $Ca^{2+}$ (N = 6 cells, red data points).

The following figure supplements are available for figure 6:

**Figure supplement 1**. Flow diagram detailing MSD difference ratio ($R_{\Delta MSD}$) calculation from top to bottom.

**Figure supplement 2**. Mobility change in MSD calculated for 120 msec with TRPV1-GFP channels after capsaicin treatment.

---

(***Mercado et al., 2014***). The immobile TRPV1-tagRFP sparklets allowed us to measure the fluorescence intensity of capsaicin-induced sparklets and determine the number of channels at each sparklet site. The two-state fluorescence signature of the TRPV1-tagRFP sparklets could arise from single-channel openings or from multiple channels at each sparklet site with very high gating cooperativity (***Figure 2***). Analysis of the photobleaching steps attributed to TRPV1-GFP channels in fixed HEK293T/17 cells provided further confirmation that the channel assembles as a tetramer of TRPV1-GFP subunits without forming higher order oligomeric assemblies (***Figure 3***). Importantly, laterally mobile TRPV1-GFP in live cells could be tracked via both their fluorescent protein label and their sparklet activity, and switching from a lower fluorescence emission of the fluorescent protein label to the higher intensity sparklet fluorescence within the same feature demonstrated that the mobile sparklets emanated from mobile TRPV1-GFP (***Figure 5*** and ***Video 9*** and ***Video 10***). Together, these data indicate that each capsaicin-induced sparklet is due to the activation of one, and only one, TRPV1 ion channel. This doesn't preclude higher number assemblies from forming, but these were never observed in our transiently transfected cells. Our data on dorsal root ganglion sparklets activated by capsaicin recapitulates our observations in TRPV1-GFP and TRPV1-tagRFP expressing HEK293T/17 cells (***Figure 1A*** and ***Figure 1—figure supplement 1***).

Our investigation into the lateral mobility of TRPV1 as it is influenced by the channel's activity has barely begun to scratch the surface of what may be an important new mode of functional regulation. However, our finding that $Ca^{2+}$ influx is required for the capsaicin-induced decrease in TRPV1 mobility

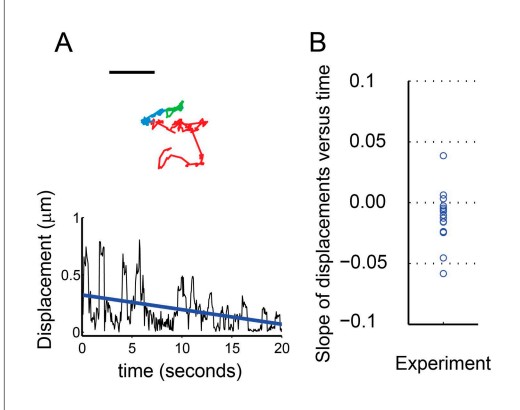

**Figure 7**. Characterization of mobile TRPV1 sparklets in voltage-clamped HEK293T/17 cells. (**A**) Trajectory of sparklet with decreasing mobility. The total steps of the trajectory are divided into three equal parts representing the first third (red), the second third (blue) and the last third (green) of each track. Scale bar is 1 μm. Displacement (see 'Materials and methods') plot of sparklet in (**A**) is shown below. The blue line is the slope used to characterize the change in displacements over time. (**B**) Slopes corresponding to the change in displacements of mobile sparklets (N = 16) in density plot with the center of the distribution located below 0.0.

The following figure supplement is available for figure 7:

**Figure supplement 1**. Characterization of mobile TRPV1 sparklets in voltage-clamped HEK293T/17 cells.

may hint at the mechanism by which the change in mobility occurs. Changes in the cytoskeleton upon upregulation of $Ca^{2+}$-sensitive effectors such as α-actinin, gelsolin or scinderin (**Rodriguez Del Castillo et al., 1990**; **Witke et al., 1993**; **Kinosian et al., 1998**) can be easily imagined to regulate membrane protein mobility. The formation of local $Ca^{2+}$ domains has previously been linked to regulation of cell architecture and remodeling of the cytoskeleton. A recent study looking at the influence of TRPM7 elicited $Ca^{2+}$ micro-domains in migrating fibroblasts identified this process as part of the mechanism by which the cell steers towards a chemoattractant (**Wei et al., 2009**, **2012**). Thus, the involvement of cytoskeleton in the $Ca^{2+}$-influx mediated regulation of TRPV1 mobility is a subject of interest for future studies.

We and others have shown that TRPV1 binds to several proteins important to cell signaling, including the p85β subunit of PI3-kinase and the TrkA receptor (**Chuang et al., 2001**; **Stein et al., 2006**). These interactions are likely related to a physiological response facilitated by nerve growth factor (NGF) that increases cell surface expression of TRPV1 (**Zhang et al., 2005**; **Stein et al., 2006**). Biochemical and electrophysiological experiments also point to a complex formed between AKAP79/150 and TRPV1 to alter sensitization of TRPV1 in response to inflammatory agents (**Zhang et al., 2008**; **Por et al., 2010**; **Efendiev et al., 2013**). Differential properties of a signaling complex involving TRPV1 could be determined by the mobility of the channel, which makes its activity dependent mobility a point of interest. One can imagine that the activity of TRPV1 will confine the channel to a specific region of the cell, where it may (or may not) be more likely to encounter its binding partners in the PI3-kinase/ TrkA receptor complex. It is further possible that the binding of TRPV1 to a complex with a consequent change to the channel's mobility also feeds back on its function.

At this point, we can only speculate on the physiological role of activity-regulated changes in TRPV1 mobility. Perhaps decreased mobility in regions of locally elevated channel activity serves as a sort of diffusion trap, concentrating TRPV1 channels in parts of the cell where they are most needed. Sensory neurons with cell bodies in the dorsal root ganglia are not very polarized (**Dubin and Patapoutian, 2010**). Indeed, in culture without other types of neurons they do not form synapses; in their role as sensory receptors they are not postsynaptic to other cells types and appear to have lost the signaling required to form postsynaptic structures. Other mechanisms for protein targeting might thus be prominent in these types of cells.

## Materials and methods

Molecular Cloning—The C-terminal GFP-labeled and tagRFP-labeled TRPV1 were made by subcloning the relevant fragments into the pCDNA3 vector, as previously described for our TRPV1-CFP (cyan fluorescent protein) construct (**Ufret-Vincenty et al., 2011**).

Cell Culture and Electrophysiology—Mouse dorsal root ganglia cells were isolated as previously described (**Stein et al., 2006**). Nerve growth factor was added to the final medium at a concentration of 100 ng/ml and experiments were completed within 24 hr of sacrifice of the mouse. Whole-cell voltage clamp recordings of capsaicin dose responses in **Figure 1—figure supplement 2** were done as previously described (**Stein et al., 2006**). F11 cells were cultured as described previously

(*Stein et al., 2006*). HEK293T/17 cells (ATCC, Manassas, VA) were cultured in Dulbecco's Modified Eagle Medium (DMEM; Life Technologies) supplemented with 10% fetal bovine serum, 50 units/ml penicillin, and 50 µg/ml streptomycin at 37°C and 5% $CO_2$. For both cell lines, low expression levels of the TRPV1-GFP or TRPV1-tagRFP protein was obtained with transient transfection by Lipofectamine 2000 (Life Technologies) at suboptimal concentrations of plasmid DNA (250 ng DNA to 1 µl Lipofectamine in one 35 mm well) and decreased incubation times (2 hr) as compared to the manufacturer's instructions. Cells were passaged after 12 hr onto poly-L-lysine coated glass coverslips in Hank's Buffered Saline Solution (HBSS: 140 mM NaCl, 4 mM KCl, 1 mM $MgCl_2$, 1.8 mM $CaCl_2$, 10 mM Hepes, 5 mM glucose at pH 7.4) and experiments were conducted 2 hr later. Imaging of TRPV1-GFP in *Figure 4*, *Figure 6*, *Figure 6—figure supplement 2*, and *Video 8* was performed in HBSS or nominally $Ca^{2+}$-free solution (identical to HBSS except $MgCl_2$ = 2.0 mM and no added $Ca^{2+}$) as indicated. Activation of TRPV1-GFP in *Figure 6* experiments was accomplished with 1 µM capsaicin (from DMSO stock) in HBSS or nominally $Ca^{2+}$-free solution.

Ca²⁺ influx through TRPV1-GFP, TRPV1-tagRFP or native TRPV1 channels was recorded as a current in whole-cell configuration of patch clamp with the membrane potential held at −40 mV, while simultaneously observing sparklets as previously described (*Navedo et al., 2005*). During sparklet experiments in dorsal root ganglion cells (*Figure 1* and *Figure 1—figure supplement 1* and *Videos 1–3*) and moving sparklet experiments in HEK293T/17 cells (*Figure 5*, *Figure 7* and *Figure 7—figure supplement 1*, *Videos 9 and 10*), we perfused the following bath solution (in mM): 140 N-methyl-D-glucamine (NMDG), 5 CsCl, 1 $MgCl_2$, 10 glucose, 10 HEPES, and 2 $CaCl_2$ adjusted to pH 7.4, which is close to a physiological concentration of $Ca^{2+}$. In experiments quantifying the immobile sparklet intensities of TRPV1-tagRFP in HEK293T/17 cultured cells (*Figure 2* and *Figure 2—figure supplement 1*, *Video 7*), the bath solution was changed to 20 mM $CaCl_2$ and 120 mM NMDG to increase fluorescence signal (see reason given below for Fluo-5F use). In *Figure 2A* a nominally $Ca^{2+}$-free NMDG solution was used as previously described with 120 mM NMDG but with no added $Ca^{2+}$. Pipettes were filled with a solution composed of (in mM) 110 Cs-aspartate, 20 CsCl, 1 $MgCl_2$, 5 MgATP, 10 HEPES, 10 EGTA, and either 0.2 Fluo-5F (immobile sparklets in HEK293T/17 cells; *Figure 2*, *Figure 2—figure supplement 1*, *Video 7*) or Fluo-4 (dorsal root ganglion sparklets: *Figure 1* and *Figure 1—figure supplement 1* and *Videos 1–3*; moving sparklets in HEK293T/17 cells: *Figures 5 and 7*, *Figure 7—figure supplement 1*, *Videos 9 and 10*) adjusted to pH 7.2 with CsOH. Both Fluo-4 and Fluo-5F were purchased from Life Technologies. Although Fluo-4 was used in dorsal root ganglion cell experiments and moving sparklet experiments in HEK293T/17 cells because of its increased affinity for $Ca^{2+}$ and improved fluorescence signal in physiological solutions (2 mM $Ca^{2+}$), we used Fluo-5F dye for the immobile sparklet experiments in HEK293T/17 because we wished to more closely follow the methods of *Navedo et al. (2005)*, who previously studied $Ca^{2+}$ sparklets from voltage-gated L-type $Ca^{2+}$ channels in HEK293T/17 cells, and the cells were also allowed to equilibrate with the pipette contents for 2 min before each experiment. For F-11 cell sparklet experiments (*Figure 1—figure supplement 3* and *Videos 4 and 5*), we used standard HBSS buffer and 0.2 mM Fluo-4 pipette solution, as described above. TRPV1 activation was induced by perfusion with either of the previously mentioned NMDG bath solutions supplemented with 100 nM capsaicin (from DMSO stock). For some experiments, $Ca^{2+}$ was depleted from intracellular stores by pre-incubating the cells for 5 min before experiments commenced in the NMDG buffer supplemented with 1 µM thapsigargin.

Total internal reflection fluorescence (TIRF) microscopy—Fluorescence images of cellular structure in close proximity to the coverslip (~100 nm) were obtained through a Nikon Ti TIRF illumination microscope with a 100× 1.45 NA oil objective and a QuanEM:512SC Photometrics electron multiplying CCD camera. The 488 nm line of a 100 mW argon laser (Melles Griot, Albuquerque, NM) was used for all fluorophores except tagRFP, for which a 40 mW 561 nm solid state laser (Coherent, Santa Clara, CA) was used. Imaging of TRPV1-GFP for channel tracking (*Figures 4 and 6*, *Figure 6—figure supplement 2*, *Video 8*) was done at 33 Hz at 100% laser power. Mobile sparklets were captured at 33 Hz at 100% laser power while immobile sparklet intensity experiments (except experiments in *Figure 1A* and *Figure 1—figure supplement 1*) were recorded at 50 Hz at 20% laser power. All sparklet imaging began with capsaicin perfusion. Raw images acquired with NIS Elements (Nikon Instruments Inc., Melville, NY) were imported into ImageJ (http://rsbweb.nih.gov/ij/) for post-acquisition processing.

Image Processing and Analysis—Data collected on moving TRPV1-GFP (*Figure 4*) was processed in ImageJ by 1) removing background with a three pixel radius rolling ball filter, 2) averaging five frames together with the Windowed-sinc Filter plugin (http://rsb.info.nih.gov/ij/plugins/windowed-sinc-filter.html)

using a spatial frequency cut-off of 0.2, and 3) filtering the fluorescent feature with the spot enhance plug-in at 1.25 pixels (Daniel Sage, spot tracker 2D, *Sage et al., 2005*). Detection and tracking of the channels was accomplished with the u-track software (http://lccb.hms.harvard.edu/software.html) in Matlab (www.Mathworks.com, Natick, MA) (*Jaqaman et al., 2008*). Within the u-track program, the following features were selected: We chose to supply a background sample for detection purposes, the maximum gap between two segments of a track could be eight frames, the option to search for directed movement was deselected, and a maximum of three pixels movement between frames was imposed as a rule for finding tracks.

Mean squared displacements [(MSD), $\left\langle \left( \vec{R}(t+\tau) - \vec{R}(t) \right)^2 \right\rangle$] in *Figure 4* were calculated from time averages of the x and y coordinates determined in the aforementioned u-track program. To estimate an effective diffusion, MSD values for lag times of up to 10 frames (1 frame = 30 msec) were fitted with the equation for a particle diffusing in two dimensions [$\left\langle \Delta \vec{R}^2(t) \right\rangle = 4Dt$] (*Berg, 1993*).

In preparation for analyzing sparklet intensity and movement observed in endogenous TRPV1, TRPV1-tagRFP and TRPV1-GFP expressing cells, we first normalized TRPV1 sparklet signal to an average from images acquired before sparklets were observed, as previously described (*Demuro and Parker, 2005*). In the case of immobile sparklet intensity analysis (*Figures 1A and 2* and *Figure 1—figure supplement 1*, and *Figure 2—figure supplement 1*), all points histograms of the normalized intensity were generated in Igor Pro (Wavemetrics, Lake Oswego, OR), with baseline subtraction to account for rising global $Ca^{2+}$ signal in background and ($\Delta F/F_0$) collected in a 12 × 12 (6 × 6 for dorsal root ganglion neuron experiments) pixel region-of-interest (ROI) centered on each sparklet. In the case of sparklet movement analysis (*Figures 1B, 5, 7*) five frames were averaged in time with the frequency cut-off set to 0.1 using the Windowed-sinc filter plug-in (ImageJ). Tracking of individual sparklets was then semi-autonomously performed with the ImageJ Spot Tracker 2D plug-in (http://bigwww.epfl.ch/sage/soft/spottracker/). In *Figure 5*, the TRPV1-GFP fluorescent feature was tracked at an early stage of sparklet activity so that longer segments of the track represented contiguous positions of the GFP fluorescence followed by segments of sparklet fluorescence. Spot Tracker 2D does not provide the uncertainty for each position in the track, and we wished to compare the precision in measuring sparklet to GFP single molecule positions. We used Matlab (www.Mathworks.com) to calculate the standard deviations (x- and y-direction) of a 2D bivariate Gaussian function fit to an ROI centered on each fluoroescent peak recovered by ImageJ Spot Tracker 2D (*Schmidt et al., 1996*; *Cheezum et al., 2001*). A bivariate Gaussian was used because time averaging of the frames could stretch the feature by a small degree, and the uncertainty for the precision was calculated as the average of the two standard deviations. The region we ascribed to the feature (5 × 5 pixels) was chosen as a compromise between isolating the feature of interest from surrounding unrelated signals and recovering the full shape of the GFP or sparklet feature. As TRPV1 channels opened, the background fluorescence levels gradually increased during the course of the video. Although most single particle detection methods rely on a background offset in their fitting function to accommodate a slowly varying and evenly distributed background level, we found that the highly variable and changing background levels of sparklet videos necessitated a background subtraction and additional fitting constraints in the tails of the 2D Gaussian fit. To suppress any effect that the background fluorescence had on our fits, an additional two layers of pixels surrounding the 5 × 5 pixel region were used to calculate a mean background level. This mean background was subtracted from the pixels in our 5 × 5 feature region, and the two layers of pixels were padded onto our feature region as zero-valued to extend the total fitted ROI to a 9 × 9 region. In *Figure 7*, tracks were selected based on their contiguous sparklet fluorescence signals for ease of tracking. Fluorescence from GFP was not observed in the course of these videos due to the high background fluorescence (for example, see *Video 6*). This allowed us to pool displacement data from TRPV1-GFP and TRPV1-tagRFP in our analysis. Displacements of sparklet positions in *Figure 7* and *Figure 7—figure supplement 1* were determined for time points that are 10 steps apart (i.e., frame *i* to *i* + 10) to: 1) minimize the random noise contribution and 2) have greater confidence in the precision of the displacement calculation.

Quantification used to assess the effects of capsaicin on TRPV1-GFP mobility in *Figure 6* and *Figure 6—figure supplement 2* was based on the trajectories acquired through the u-track program as described above with the following additional steps: 1) A mask to exclude regions of high channel densities is used when deemed necessary and consistently applied to all videos made on the same cell. 2) The MSD was calculated for time points 90 msec (3 frames) apart in *Figure 6* and

120 msec (4 frames) apart in *Figure 6—figure supplement 2*. 3) An MSD difference metric to compare the mobility of channels in the same cell before and after capsaicin treatment was implemented to emphasize trajectory differences due to TRPV1-GFP movements and to minimize changes over time caused by photobleaching. The number of tracks contributing to each average MSD in a video ranged from ~80-600. The MSD difference ratio ($R_{\Delta MSD}$) reduced the impact of photobleaching on a particular cell by normalizing to the MSD measured in the first part of each video (see *Figure 6—figure supplement 1*).

Single Molecule Photobleaching—HEK293T/17 cells, transiently transfected for low expression levels of TRPV1-GFP, were transferred to uncoated coverslips and allowed to recover in HBSS for 2 hr before being fixed with a 12 min incubation in a 4% paraformaldehyde solution prepared in HBSS. HBSS buffer was also used for the rinses and for imaging. Imaging with TIRF microscopy on the previously described Nikon instrument was performed at 100% 488 nm laser power, 10 Hz acquisition speed, and with $2 \times 2$ binning of the pixels using the QuantEM camera. Post-acquisition processing of the images entailed subtracting off a $7 \times 7$ pixel smoothed image from each frame and identifying potential bleach steps in the temporal fluorescence intensity traces of $3 \times 3$ ROIs centered on individual channel sites as previously described (*Demuro et al., 2011*). A zero-truncated binomial distribution (*R-forge, 2008–2009*), $\binom{n}{k}\dfrac{p^k(1-p)^{n-k}}{1-(1-p)^n}$, was fitted to the data of photobleaching steps by maximum likelihood estimates using Matlab (www.Mathworks.com) to determine the probability of functional GFP in fluorescent features containing up to four, five, or eight GFP molecules. The best fit probability for GFP fluorescence, $p$, and number of GFP sites per fluorescent feature, $n$, was selected using a $\chi^2$ goodness-of-fit test. For ease of interpretation the fluorescence intensity traces in *Figure 3A* were also filtered with a three-point running average.

## Acknowledgements

We thank Mika Munari for expert technical assistance with the molecular biology and cloning, Dr Alexander T Stein for collecting the data shown in *Figure 1—figure supplement 2*, and Drs Manuel Navedo, Greg Horwitz and William N Zagotta for helpful discussions. Research reported in this publication was supported by the National Eye Institute of the National Institutes of Health under award number R01EY017564, by the National Institute of General Medical Sciences of the National Institutes of Health under award number R01GM100718, and by the following additional awards from NIH: S10RR025429, T32HL007312, and P30 DK017047.

## Additional information

### Funding

| Funder | Grant reference number | Author |
| --- | --- | --- |
| National Eye Institute | R01EY017564 | Sharona E Gordon |
| National Institute of General Medical Sciences | R01GM100718 | Sharona E Gordon |
| National Institutes of Health | T32HL007312 | Eric N Senning |

The funders had no role in study design, data collection and interpretation, or the decision to submit the work for publication.

### Author contributions

ENS, Conception and design, Acquisition of data, Analysis and interpretation of data, Drafting or revising the article; SEG, Conception and design, Analysis and interpretation of data, Drafting or revising the article

### Ethics

Animal experimentation: Many local labs generating transgenic mice have a surplus of wild-type littermates that are euthanized after phenotyping. We have arranged to be present during the euthanasia process and immediately take the carcass for harvesting of the tissue we require. These arrangements are consistent with the principle of reducing the total number of animals used in biomedical

experimentation and are approved by AALAC and the University of Washington IACUC. All experiments were conducted in accordance with University of Washington Animal Protocol #3190-04.

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
