## [Decision Letter]

[Editors’ note: this article was originally rejected after discussions between the reviewers, but the authors were invited to resubmit after an appeal against the decision.]

Thank you for choosing to send your work entitled “Activity and Ca^2+^ regulate the mobility of TRPV1 channels in the plasma membrane of sensory neurons” for consideration at *eLife*. Your full submission has been evaluated by John Kuriyan (Senior editor), Rick Aldrich (Reviewing editor), and two other peer reviewers. The decision was reached after discussions between the reviewers. We regret to inform you that your work will not be considered further for publication.

All the reviewers recognize the work as an impressive and interesting technical achievement. However they feel that a substantial amount of additional work would be necessary to fully address the technical issues delineated in the individual reviews, which are appended below. *eLife* seeks to avoid requiring authors to preform extensive additional experimental work as a requirement for acceptance.

*Reviewer #1*:

The work is a technical achievement in simultaneously tracking movement of TRPV1 channels in the membrane and channel activity, measured by calcium sparklets. The central interesting finding is that channel mobility decreases after activation by capsaicin. The data are nice and the story is well presented.

There is a serious deficit is in the statistical treatment of the photo bleaching data. A “best fit” is presented with no description of how it was determined to be best or how much better it was than fits with other parameters.

*Reviewer #2*:

In this paper by Senning and Gordon, the diffusional behavior of the ion channel TRPV1 at the plasma membrane is studied with TIRF microscopy in DRG neurons, F11 and HEK cells. The authors succeed in imaging the real–time activity of the channel by imaging calcium “sparklets” that occur as the moving channel gates. The authors should be commended in pioneering and accomplishing this difficult experiment. In general, I think this paper is interesting and opens up new and exciting directions for ion channel research. The main question of the paper is summarized in the authors’ own words: “whether actively gating channels can be mobile and whether their mobility is linked to their activity.” This has been an unanswered question. How mobility changes in a channel might affect signaling, however, is still unknown. The data in the paper is well presented and the findings are clear. The paper is well written. I have several comments that the authors should address.

1) As far as I can tell from the data the majority of the imaging was done in patch-clamped cells in the presence of cytosolic calcium chelator and thapsigargin. Did the authors observe the same diffusional behavior and change in diffusion with the addition of calcium and ligand in intact (non–patched) cells expressing TRPV1–GFP (or RED)? I am concerned that the chemical manipulations that aided in imaging sparklets could change the “wild–type” diffusional behavior of the channel? You could imagine that if the cytoskeleton is altered by calcium (which it is) and the channel's movements are dependent on the cytoskeleton then these manipulations which buffer calcium could have non–physiological consequences on the channel's behavior. Showing data of TRPV1–GFP in intact cells (with and without ligand) would be a helpful control for this concern.

2) Does every channel gate (have a sparklet) or is there only a subset of channels that do? Do all channels slow down or just a subset? For example, do only channels with sparklets slow or do all channels slow regardless of their activity? It is difficult to see this from the way the data is presented. Specifically, the data is plotted as means from individual cells but in this format I cannot tell if all the channels slow or just a few have massive changes in diffusion. A supplement with histograms of diffusions for all tracked channels or box–and–whisker plots of the full data set would be nice. Is the behavior homogeneous across the cell or are there regional differences in diffusion across the foot–print?

3) Is this a general phenomenon for all membrane proteins (or ion channels) or is this specific for TRPV1? Do other ion channels show this behavior in these cells under these conditions? Is this effect due to a global change in the cell, membrane, or cytoskeleton?

4) How does the duration of the sparklet (visible open time) compare to single channel recordings of TRPV1? The open times seem very long (∼10 sec). Is this seen in electrophysiology data? The authors should discuss in more detail how their data compares to past recordings of TRPV1 under similar conditions?

5) How does the diffusion coefficient of TRPV1 compare to other membrane proteins or lipids? Could the authors please discuss this in more detail the paper?

*Reviewer #3*:

The authors provide the first measurements of elementary calcium fluxes through TRPV1 channels, and perform measurements of TRPV1 motility with single particle tracking. These are very demanding experiments, and individually intrinsically difficult to analyze. Together, it presents an exceptionally difficult challenge. The authors present evidence that TRPV1 channel activity may affect channel motility. The current work lacks a coherent rationale, experimental structure, mechanistic basis, and physiological relevance. The study seems to be an assortment of different control experiments with some superficial analysis.

Concerns with Ca^2+^ imaging:

Experiments were performed similar to how Santana and colleagues performed their L–type Ca^2+^ channel sparklets. With fluo–4 and EGTA (10 mM) delivered through a patch pipette, and cells held at a negative holding potential. The only difference is that they held at –40 mV (compared to –70 mV for L–type sparklets). They argued that this was so “to minimize the activity of voltage–gated Ca^2+^ channels while also limiting the driving force of Ca^2+^ into the cell, which would reduce the duration of our experiments as the global rise in Ca^2+^ obscures the signal due to single TRPV1”. So it isn't clear why –40 mV was chosen, L–type calcium channel activity would likely be greater at –40 mV than –70 mV. Also –40 mV does not minimize the driving force when the calcium equilibrium potential is +120 mV. In addition, the 10 mM EGTA should be capable of buffering global changes in cytosolic Ca^2+^. Furthermore, these recordings represent the first measurements of TRPV1 sparklets, and therefore a number of criteria should be fulfilled, e.g., quantal levels should depend on the calcium electrochemical gradient and not on the nature of the agonist or antagonist. TRPV1 agonists and antagonists should affect the probability of a sparklet but not the quantal level.

Concerns with single particle tracking:

The power of single particle tracking analysis relies on the algorithms used to define and measure the displacement of a fluorescent pixel pattern (which needs to better explained in the manuscript). Typically, the location of an individual particle is defined by the fitting a 2D Gaussian function with a set standard deviation that encompasses the pixel pattern. These algorithms work well with single fluorescent particles due to the high signal–to–noise ratio between the particle and the surrounding environment (i.e., small standard deviation of the 2D Gaussian fit that equal the relative size of the particle). Like any statistical analysis, the smaller the variability, the higher the confidence in the calculated mean value, or in this case the 2D location of the particle. Attempts to apply these methods of analysis to larger more variable and dynamic “particle” like Ca^2+^ release events would be extremely difficult. The relative size, or spatial spread of Ca^2+^, would increase the standard deviation of the 2D Gaussian fit, and therefore decrease the confidence of the location of the particle. To strengthen these data, the authors would need to perform several control experiments to show that this is not true. With the ability to simultaneous record Ca^2+^ events and GFP tagged channel, do the MSD's for the Ca^2+^ event match up with the trajectory of individual tagged channels?

Concerns with final interpretation:

How do the authors resolve the movement of a single particle of GFP during a Ca^2+^ event when the fluorescence signal of a given area is saturated with the underlying fluo–4 signal? I would predict the standard deviation of 2D Gaussian fit would increase, therefore decreasing the confidence of the 2D mean placement. This may lead to the interpretation that the mobility of the channel to be slower (or no not moving at all).

What is the density of TRPV1 sparklet sites for a given cell? It is apparent from Figure 2, that the number of channels on the plasma membrane is high (Methods: ∼80–600 channels), yet not all of those channels may be actively conducting Ca^2+^ to elicit a sparklet (it would be hard spatially resolve 80–600 individual sparklet sites). Therefore, the analysis performed of Figure 7 may not accurately characterize the Ca^2+^ dependency of channel mobility as it cannot separate the active from the inactive channels.

In the Results section, the authors conclude that TRPV1 channels exhibited a decrease in mobility following that activation of Ca^2+^ sparklets by capsaicin. With the current data, it would be difficult to conclude whether the slower mobility occurs before or after the influx of Ca^2+^. This is the old chicken versus the egg conundrum. One possible way to address this would be to plot the Ca^2+^ event (*F/F*_*0*_) and time–averaged MSDs on the same time scale, and see whether the Ca^2+^ event coincides with a change in the MSD for a given site.

As indicated in the Discussion, there is evidence that TRPV1 and AKAP79/150 form a complex which presumably would affect motility as well as the activation of associated calcineurin. Does either of the cell types used here (DRG or HEK cells) express AKAP150, and is it related to motility?

---

## [Author Response]

The reviewers expressed considerable enthusiasm for the work, but noted that extensive changes would be required. We have taken the critiques very seriously, and we have performed both new experiments and considerable new analysis that we believe would fully satisfy the reviewers. Our observations of gating events in diffusing ion channels is unprecedented. Our finding that mobility is coupled to channel activity and Ca^2+^ influx open a whole new field of study.

Reviewer #1:

*The work is a technical achievement in simultaneously tracking movement of TRPV1 channels in the membrane and channel activity, measured by calcium sparklets. The central interesting finding is that channel mobility decreases after activation by capsaicin. The data are nice and the story is well presented*.

*There is a serious deficit is in the statistical treatment of the photo bleaching data. A “best fit” is presented with no description of how it was determined to be best or how much better it was than fits with other parameters*.

We have included new analysis of the data, fitting them with alternative models and showing the results of these fits in new Figure 3. The statistical analysis we used to discriminate among the models and the confidence with which we can discriminate are now fully described. Specifically, with *n*=4 the estimate for *p*=.45 is correct with a 95% confidence interval of 0.42-0.48. Chi^2 test of the observed versus expected pdf values was 0.1712 and should not be rejected. An identical fitting treatment with *n*=5 and *n*=8 yielded Chi^2 values of 4.0717 and 16.386, respectively, and were rejected for the tabulated values with appropriate degrees of freedom.

Reviewer #2:

*In this paper by Senning and Gordon, the diffusional behavior of the ion channel TRPV1 at the plasma membrane is studied with TIRF microscopy in DRG neurons, F11 and HEK cells. The authors succeed in imaging the real–time activity of the channel by imaging calcium “sparklets” that occur as the moving channel gates. The authors should be commended in pioneering and accomplishing this difficult experiment. In general, I think this paper is interesting and opens up new and exciting directions for ion channel research. The main question of the paper is summarized in the authors’ own words: “whether actively gating channels can be mobile and whether their mobility is linked to their activity.” This has been an unanswered question. How mobility changes in a channel might affect signaling, however, is still unknown. The data in the paper is well presented and the findings are clear. The paper is well written. I have several comments that the authors should address*.

*1) As far as I can tell from the data the majority of the imaging was done in patch-clamped cells in the presence of cytosolic calcium chelator and thapsigargin. Did the authors observe the same diffusional behavior and change in diffusion with the addition of calcium and ligand in intact (non–patched) cells expressing TRPV1–GFP (or RED)? I am concerned that the chemical manipulations that aided in imaging sparklets could change the “wild–type” diffusional behavior of the channel? You could imagine that if the cytoskeleton is altered by calcium (which it is) and the channel's movements are dependent on the cytoskeleton then these manipulations which buffer calcium could have non–physiological consequences on the channel's behavior. Showing data of TRPV1–GFP in intact cells (with and without ligand) would be a helpful control for this concern*.

The reviewer’s comments brought to our attention a serious problem with the clarity of our presentation. We had exactly the concerns raised by the reviewer, and we included the appropriate controls in the original submission of the manuscript. We have significantly improved the presentation by reorganizing the order of the presentation, more clearly stating experimental conditions in the text, and adding needed information to the figure legends. The requested studies of TRPV1–GFP mobility in non-patch clamped cells allowed to regulate their Ca^2+^ levels without interference of experimentally added Ca^2+^ buffers are shown in Figure 4 and Figure 6 and discussed in their own section of the text. In addition, we have new analysis that demonstrates parallel behavior by TRPV1–GFP in non-patched cells and TRPV1–GFP with sparklets in cells dialyzed with EGTA and a Ca^2+^ dye and studied under voltage-clamped conditions (Figure 5).

*2) Does every channel gate (have a sparklet) or is there only a subset of channels that do? Do all channels slow down or just a subset? For example, do only channels with sparklets slow or do all channels slow regardless of their activity? It is difficult to see this from the way the data is presented. Specifically, the data is plotted as means from individual cells but in this format I cannot tell if all the channels slow or just a few have massive changes in diffusion. A supplement with histograms of diffusions for all tracked channels or box–and–whisker plots of the full data set would be nice. Is the behavior homogeneous across the cell or are there regional differences in diffusion across the foot–print*?

The relationship between activity and mobility for single TRPV1 molecules is not a question that would have been raised prior to this study. We agree with the reviewer that, in light of our new work, this relationship begs to be addressed. We are currently developing the technology that will be required to address the reviewer’s questions, but we believe that the set of required experiments represents a full R01 proposal, well beyond the scope of the current study. Independent color channels will have to be used for sparklets and TRPV1, so that mobility and activity can be independently measured. Nonetheless, we provide new analysis in new Figure 5 to move as close to this question as possible using existing methods. We have also included new experiments to allow the behavior of individual sparklets to be compared across different types of cells, and show these data in new Figure 2 and new Figure 2.

*3) Is this a general phenomenon for all membrane proteins (or ion channels) or is this specific for TRPV1? Do other ion channels show this behavior in these cells under these conditions? Is this effect due to a global change in the cell, membrane, or cytoskeleton*?

We share the reviewer’s excitement to learn whether the new phenomenon we present here represents a general cellular strategy for controlling activity and mobility. Although a survey of many membrane proteins using our approach would be useful, the literature already provides us with a partial answer. Sparklets have been reported for L–type Ca^2+^ channels, AchR channels, and TRPV4 channels. In all reported cases, the sparklets were not mobile. Yet TRPV1 is mobile in at least three cell types: isolated dorsal root ganglion neurons, HEK293T/17 cells, and F11 cells. We believe mobile gating channels are likely to be the rule, rather than the exception, but it will take many studies by a number of groups to establish whether this is the case.

*4) How does the duration of the sparklet (visible open time) compare to single channel recordings of TRPV1? The open times seem very long (∼10 sec). Is this seen in electrophysiology data? The authors should discuss in more detail how their data compares to past recordings of TRPV1 under similar conditions*?

We have performed the requested analysis and include the results in new Figure 1, Figure 1—figure supplement 2, and Figure 2. We have used these figures as the basis for new discussion in the main text.

*5) How does the diffusion coefficient of TRPV1 compare to other membrane proteins or lipids? Could the authors please discuss this in more detail the paper*?

We now show that the variance of TRPV1 mobility within a given cell is at least an order of magnitude greater than that seen between cells. We have improved the Discussion to put the effective diffusion constants measured for TRPV1 in a broader context.

Reviewer #3:

*The authors provide the first measurements of elementary calcium fluxes through TRPV1 channels, and perform measurements of TRPV1 motility with single particle tracking. These are very demanding experiments, and individually intrinsically difficult to analyze. Together, it presents an exceptionally difficult challenge. The authors present evidence that TRPV1 channel activity may affect channel motility. The current work lacks a coherent rationale, experimental structure, mechanistic basis, and physiological relevance. The study seems to be an assortment of different control experiments with some superficial analysis*.

*Concerns with Ca*^*2+*^
*imaging*:

*Experiments were performed similar to how Santana and colleagues performed their L–type Ca*^*2+*^
*channel sparklets. With fluo–4 and EGTA (10 mM) delivered through a patch pipette, and cells held at a negative holding potential. The only difference is that they held at –40 mV (compared to –70 mV for L–type sparklets). They argued that this was so “to minimize the activity of voltage–gated Ca*^*2+*^
*channels while also limiting the driving force of Ca*^*2+*^
*into the cell, which would reduce the duration of our experiments as the global rise in Ca*^*2+*^
*obscures the signal due to single TRPV1”. So it isn't clear why –40 mV was chosen, L–type calcium channel activity would likely be greater at –40 mV than –70 mV. Also –40 mV does not minimize the driving force when the calcium equilibrium potential is +120 mV*.

We thank the reviewer for alerting us to difficult-to-follow explanation in the original manuscript. We chose the holding potential for our experiments empirically, based on our recordings of isolated mouse dorsal root ganglion neurons. Using the same recording conditions as those used to examine TRPV1, we imaged Fluo–4 with 10 mM EGTA, but no capsaicin, at a number of different holding potentials. We found that holding at –40 mV produced the smallest rise in intracellular Ca^2+^ over the ten-or-so minutes needed for our experiments. We have now explained that more clearly in the text. Because isolated neurons have a host of natively expressed channels and we were able to identify conditions in which the Ca^2+^ leak was minimal, we did not attempt to assign the background rise in Ca^2+^ to any particular type of channel.

*In addition, the 10 mM EGTA should be capable of buffering global changes in cytosolic Ca*^*2+*^*. Furthermore, these recordings represent the first measurements of TRPV1 sparklets, and therefore a number of criteria should be fulfilled, e.g., quantal levels should depend on the calcium electrochemical gradient and not on the nature of the agonist or antagonist. TRPV1 agonists and antagonists should affect the probability of a sparklet but not the quantal level*.

We found that 10 mM EGTA was indeed very effective at increasing the duration of our experiments, and it is critical to observing micro-domains of Ca^2+^ sparklets since the on rate of Fluo–4 is higher than the on rate of EGTA, resulting in a localization and improved kinetic resolution of Ca^2+^ sparklet signal (Shuai and Parker, 2005). However, with extracellular Ca^2+^ at 2 mM, the high Ca^2+^ influx through TRPV1 will continue to saturate available EGTA and a Ca^2+^ gradient will form between the plasma membrane (where we collect our signal in TIRF) and the opening of the pipette (where Ca^2+^-bound EGTA diffuses out of the cell as Ca^2+^ –free EGTA diffuses in). This increases the free Ca^2+^ available to bind Fluo–4 or Fluo-5F, making it harder to resolve additional sparklets. In response to the reviewer’s question about quantal level, we now show collected data for sparklets from both dorsal root ganglion neurons and HEK293T/17 cells in new Figure 2. The results of this analysis surprised us, because we expected the uneven position of the bilayer in the evanescent field would produce more variability than observed. The reviewer’s reasoning is sound, and now discussed in the text.

*Concerns with single particle tracking*:

*The power of single particle tracking analysis relies on the algorithms used to define and measure the displacement of a fluorescent pixel pattern (which needs to better explained in the manuscript). Typically, the location of an individual particle is defined by the fitting a 2D Gaussian function with a set standard deviation that encompasses the pixel pattern. These algorithms work well with single fluorescent particles due to the high signal–to–noise ratio between the particle and the surrounding environment (i.e., small standard deviation of the 2D Gaussian fit that equal the relative size of the particle). Like any statistical analysis, the smaller the variability, the higher the confidence in the calculated mean value, or in this case the 2D location of the particle. Attempts to apply these methods of analysis to larger more variable and dynamic “particle” like Ca*^*2+*^
*release events would be extremely difficult. The relative size, or spatial spread of Ca*^*2+*^*, would increase the standard deviation of the 2D Gaussian fit, and therefore decrease the confidence of the location of the particle. To strengthen these data, the authors would need to perform several control experiments to show that this is not true. With the ability to simultaneous record Ca*^*2+*^
*events and GFP tagged channel, do the MSD's for the Ca*^*2+*^
*event match up with the trajectory of individual tagged channels*?

*Concerns with final interpretation*:

*How do the authors resolve the movement of a single particle of GFP during a Ca*^*2+*^
*event when the fluorescence signal of a given area is saturated with the underlying fluo–4 signal? I would predict the standard deviation of 2D Gaussian fit would increase, therefore decreasing the confidence of the 2D mean placement. This may lead to the interpretation that the mobility of the channel to be slower (or no not moving at all)*.

We have done as the reviewer requested and improved the explanation of the tracking algorithm used in the text. Furthermore, we have analyzed the width of the Gaussian used to fit the sparklet/GFP position at each point, and used this value as a scaling factor for the size of the circles shown in new Figure 5. This new analysis shows that the brighter fluorescence observed with sparklets compared to GFP did not reduce our ability to precisely determine the position of sparklets (uncertainty of ∼200-300 nm) within the context of a, relatively speaking, larger TRPV1 sparklet track ( ∼5-10 μm length). Thus, although the width of sparklet Gaussians was generally larger than that of Gaussians used to localize TRPV1- GFP, we were nonetheless able to combine GFP and sparklet localizations to plot single channel trajectories. Moreover, the mobility of sparklets analyzed in Figure 7 relies on the fluorescence of sparklets and not the fluorescent protein, which permits us to determine the change in displacements over the course of the movie. We believe this new analysis significantly strengthens the manuscript, and we thank the reviewer for his/her suggestion.

*What is the density of TRPV1 sparklet sites for a given cell? It is apparent from*
Figure 2*, that the number of channels on the plasma membrane is high (Methods: ∼80–600 channels), yet not all of those channels may be actively conducting Ca*^*2+*^
*to elicit a sparklet (it would be hard spatially resolve 80–600 individual sparklet sites). Therefore, the analysis performed of*
Figure 7
*may not accurately characterize the Ca*^*2+*^
*dependency of channel mobility as it cannot separate the active from the inactive channels*.

See the response to Reviewer #2, question 2. Also, the activity- and Ca^2+^-dependent reduction in TRPV1 mobility was observed by following TRPV1–GFP in (new) Figure 6 as well as by following sparklets in (new) Figure 7.

*In the Results section, the authors conclude that TRPV1 channels exhibited a decrease in mobility following that activation of Ca*^*2+*^
*sparklets by capsaicin. With the current data, it would be difficult to conclude whether the slower mobility occurs before or after the influx of Ca*^*2+*^*. This is the old chicken versus the egg conundrum. One possible way to address this would be to plot the Ca*^*2+*^
*event (*F/F_0_*) and time–averaged MSDs on the same time scale, and see whether the Ca*^*2+*^
*event coincides with a change in the MSD for a given site*.

How well the mobility of a single TRPV1 channel tracks its activity is indeed a question of great importance. Although current methods do not allow us to directly address the mechanism by which Ca^2+^ influx through TRPV1 decreases TRPV1 mobility, we can answer the specific question posed by this reviewer. By recording the mobility of TRPV1-GFP in cells that are not voltage clamped and in which no Ca^2+^ dye is used, we can compare capsaicin in the absence of Ca^2+^ to capsaicin in the presence of Ca^2+^. As shown in Figure 6, capsaicin only causes a decrease in TRPV1 mobility in the presence of extracellular Ca^2+^ (2 mM).